# Comparative Analysis of HPA-Axis Dysregulation and Dynamic Molecular Mechanisms in Acute Versus Chronic Social Defeat Stress

**DOI:** 10.3390/ijms26136063

**Published:** 2025-06-24

**Authors:** Jiajun Yang, Yifei Jia, Ting Guo, Siqi Zhang, Jin Huang, Huiling Lu, Leyi Li, Jiahao Xu, Gefei Liu, Ke Xiao

**Affiliations:** 1The Brain Cognition and Brain Disease Institute, Shenzhen Institutes of Advanced Technology, Chinese Academy of Sciences, Shenzhen 518055, China; jj.yang3@siat.ac.cn; 2State Key Laboratory of Agricultural Microbiology, Huazhong Agricultural University, Wuhan 430070, China; jiayifei12@outlook.com (Y.J.); t.guo1@siat.ac.cn (T.G.);; 3Faculty of Life and Health Sciences, Shenzhen University of Advanced Technology (SUAT), Chinese Academy of Sciences, Shenzhen 518107, China

**Keywords:** acute social defeat stress, chronic social defeat stress, HPA axis, social avoidance, neuroendocrine–immune crosstalk

## Abstract

Organisms respond to environmental stress primarily through the autonomic nervous system and hypothalamic–pituitary–adrenal (HPA) axis, regulating metabolism, psychological states, and immune function and modulating memory, reward processing, and immune responses. The HPA axis plays a central role in stress response, exhibiting distinct activation patterns under acute versus chronic social defeat stress. However, differences in physiological impacts and regulatory pathways between these stress conditions remain understudied. This study integrates RNA sequencing and behavioral analyses to reveal that acute social defeat stress triggers transient anxiety-like behaviors, accompanied by systemic inflammation and immediate-early gene (IEG) activation. In contrast, chronic social defeat stress induces long-term behavioral and physiological alterations, including neurotransmitter imbalance (e.g., reduced GABA and increased glutamate), sustained activation of maladaptive pathways (e.g., IL-17 signaling), and disrupted corticosterone synthesis. These findings highlight the dynamic regulatory role of the HPA axis under varying stress conditions, providing novel insights into mental health disorders such as anxiety and depression. The study identifies potential therapeutic targets to mitigate chronic social defeat stress effects and offers a theoretical foundation for personalized interventions.

## 1. Introduction

Stress, an inevitable aspect of life, exerts profound effects on physiology and behavior. The duration and pattern of stress exposure dictate fundamentally divergent biological outcomes [1,2]. While acute stress responses are essential for survival, dysregulation induced by chronic stress is a major risk factor for debilitating psychiatric disorders, including disorder and anxiety [3]. Currently, several well-established stress model systems have been developed, including chronic social defeat stress (CSDS) [4,5], chronic unpredictable mild stress (CUMS) [6,7,8,9], chronic restraint stress (CRS) [10,11,12,13,14], and the learned helplessness model [15,16]. These are widely used to investigate the impact of stress on the physiology, immunity, and behavior of rodents. The CSDS model stands out due to its direct simulation of social conflict through “resident–intruder” interactions, effectively inducing behavioral phenotypes similar to human anxiety and depression, with high modeling success rates and prolonged social avoidance, making it a classic paradigm for studying stress-related mental disorders [5,17].

Psychosocial stress is known to induce abnormal immune responses within the central nervous system [18]. This includes elevated levels of pro-inflammatory cytokine *Il-1β* mRNA expression in the mouse brain and spinal cord, as well as increased corticotrophin-releasing hormone (CRH) mRNA expression in the hypothalamus and elevated plasma corticosterone concentrations [19]. Additionally, psychosocial stress triggers the generation of myeloid progenitor cells marked by CD11$b^{+}$ and Ly6$C^{\mathrm{high}}$ and leads to the accumulation of macrophages (CD11$b^{+}$ and CD4$5^{\mathrm{high}}$) in the central nervous system, alongside increased IL-6 levels [20]. As a pro-inflammatory cytokine, the increase in IL-6 levels within the central nervous system may also be associated with reduced synthesis of endothelial tight junction proteins, such as claudin-5 and occludin [21,22], which compromises the integrity of the blood–brain barrier. This, in turn, contributes to various psychological abnormalities or disorders, such as depression and anxiety. Furthermore, chronic psychosocial stress promotes an increase in glucocorticoid (GC)-insensitive monocyte populations. Recent studies have also observed that these monocytes possess the ability to transmigrate across the blood–brain barrier, potentially contributing to the development of long-term anxiety-related behaviors [23].

The hypothalamic–pituitary–adrenal (HPA) axis is the central coordinator of the stress response [24]. Upon encountering physiological stress, the HPA axis is rapidly activated, triggering the release of corticosterone (CORT), which modulates metabolic processes, immune–inflammatory responses, and psychological states [25]. Acute stress triggers a rapid physiological response by activating the hypothalamic–pituitary–adrenal (HPA) axis, resulting in a sharp increase in circulating glucocorticoids. This response induces the swift mobilization of peripheral immune cells and facilitates the subsequent restoration of homeostasis, enhancing the organism’s capacity to cope with immediate threats through the classic “fight-or-flight” response [26,27]. In contrast, chronic stress follows a dynamic progression. While early stress may be adaptively regulated by the body, prolonged or excessive stress can overwhelm the organism’s homeostatic capacity, leading to sustained overactivation of the HPA axis [28]. This drives a state of chronic low-grade inflammation, which has been implicated in the development of various psychiatric disorders, such as depression and anxiety [29].

These divergent HPA-axis dynamics drive profoundly different downstream molecular mechanisms. Acute glucocorticoid (GC) signaling induces adaptive gene expression, whereas chronic exposure often reduces glucocorticoid sensitivity through epigenetic mechanisms [30]. This desensitization dampens the immune system’s response to cortisol, ultimately leading to elevated peripheral pro-inflammatory factors (e.g., TNF, IL-17, IL-1β, IL-6) and sustained neuroinflammation, marked by chronic microglial and astrocyte activation [29,31,32,33,34]. Concurrently, chronic stress disrupts key neurotransmitter systems, alters neuronal connectivity, and disturbs synaptic network activity, leading to, for example, dysregulated excitatory glutamatergic signaling, impaired dopaminergic transmission, and altered inhibitory GABAergic signaling in specific brain regions [4,24,29,35,36,37].

Elucidating the divergent molecular response mechanisms of the HPA axis to acute versus chronic stress is crucial for understanding whether the stress response preserves adaptive homeostasis or progresses toward maladaptation and pathological states. To systematically dissect HPA-axis dysregulation and its dynamic interplay with downstream molecular pathways under these distinct stress paradigms, we combined behavioral assessments, physiological measurements, and bulk RNA-seq transcriptomic data to systematically explore HPA-axis dysregulation and its interplay with these dynamic molecular mechanisms across acute versus chronic stress. Specifically, we analyze dysregulation in inflammatory pathways, synaptic network modulation, and ligand–receptor interactions to elucidate how the HPA axis orchestrates crosstalk under stress conditions. These findings will advance mechanistic understanding of the neuroendocrine axis and uncover novel insights into stress-related pathophysiology, offering potential targets for therapeutic intervention. By delineating the adaptive vs. maladaptive mechanisms of stress responses, this work provides a foundational framework for future research on stress-associated diseases and the development of precision-based treatment strategies.

## 2. Results

### 2.1. The HPA Axis Plays a Crucial Role in the Body’s Response to Acute and Chronic Social Defeat Stress

To investigate the mechanisms underlying the coordination between the central nervous system and peripheral immune system in response to acute and chronic social defeat stress, we made specific modifications to the established protocols for both stress conditions [5,38]. For the acute social defeat stress model (ASDS), C57BL/6 mice were exposed to 1 h of acute social defeat stress and designated as the ASDS-1 h group. In a separate cohort, mice underwent the same 1 h acute social defeat stress protocol followed by a 23 h recovery period, and were designated as the ASDS-24 h group. For the chronic social defeat stress (CSDS) model, we utilized the classical CSDS paradigm [4]. After completing the CSDS model, mice were allowed 24 h of recovery, and designated as the CSDS group (Figure 1A).

In the acute social defeat stress 1 h group (ASDS-1 h), the mice exhibited increased social avoidance and anxiety-like behavior, as evidenced by a reduced social interaction ratio and decreased time spent in the social zone during the social interaction test (SIT) [39]. Similarly, the open field test (OFT) revealed reduced time spent in the center zone and fewer entries into the center zone [40]. However, after 23 h of recovery (ASDS-24 h), these behavioral changes resolved, with no significant differences observed between the acute social defeat stress 24 h group (ASDS-24 h) and the control group (Figure 1B,C and Appendix A). In contrast, the chronic social defeat stress (CSDS) group exhibited persistent behavioral changes, with significant reductions in social interaction indices, time spent in the social zone, time spent in the center zone, and the number of center zone entries during both the SIT and OFT tests, even after 24 h of recovery (Figure 1D,E). These findings suggest that acute social defeat stress induces transient behavioral changes, while chronic social defeat stress induces long-term behavioral alterations, reflecting prolonged physiological regulation.

To investigate the role of the HPA axis in mediating acute and chronic social defeat stress responses, we utilized an adrenalectomy (ADX) model to suppress HPA-axis activity and corticosterone secretion [41]. Following a 4-day recovery post-surgery, all mice underwent an OFT to assess baseline anxiety-like behaviors. Then, both sham- and adrenalectomy-model mice were subjected to a priming stimulation paradigm. After 12 h of recovery, mice underwent a second OFT assessment, followed by downstream analyses (Figure 1F).

Serum corticosterone levels were measured post-adrenalectomy and post-priming stimulation. In the sham group, priming significantly elevated corticosterone levels, whereas in adrenalectomy-model mice, corticosterone concentrations remained lower than those in untreated control mice. Notably, priming failed to induce any increase in corticosterone levels in the adrenalectomy-model group (Figure 1G). Behavioral analyses revealed that sham mice exhibited pronounced anxiety-like phenotypes following priming, as evidenced by reduced time spent in the center zone, decreased total locomotor distance, and fewer center entries during the OFT. In contrast, adrenalectomy-model mice did not display anxiety-like behaviors post-priming, with no significant changes in OFT behavioral parameters compared to pre-priming conditions. Furthermore, the proportion of time spent in the center zone was significantly higher in adrenalectomy-model mice than in sham mice post-priming (Figure 1H,I). These findings indicate that adrenalectomy abolishes stress-induced elevations in corticosterone levels and prevents the emergence of anxiety-like behaviors.

To further explore the impact of HPA-axis suppression on immune homeostasis under priming conditions, we analyzed immune cell populations in the blood, spleen, and bone marrow of adrenalectomy-model mice. Following priming, adrenalectomy-model mice exhibited a significant increase in B lymphocytes and a concurrent reduction in monocytes in both the spleen and bone marrow relative to sham mice (Figure 1J,K). Additionally, adrenalectomy-model mice showed marked reductions in CD11$b^{+}$ granulocytes and Ly6$G^{+}$ neutrophils in the bone marrow, as well as decreased circulating Ly6$C^{\mathrm{high}}$ monocytes (Appendix A). These results suggest that adrenalectomy-model mice fail to exhibit the typical stress-induced immune cell mobilization response, which is characterized by lymphocyte suppression and myeloid cell expansion [3,17,42].

Taken together, our findings demonstrate that adrenalectomy disrupts the HPA axis-mediated regulatory mechanisms by inhibiting corticosterone release, thereby impairing the organism’s ability to mount appropriate physiological and behavioral responses to stress. These results underscore the essential role of the HPA axis in orchestrating neuroimmune interactions during both acute and chronic social defeat stress adaptation.

### 2.2. Acute Social Defeat Stress (ASDS)-Induced Molecular Alterations in the HPA Axis

To investigate the immediate molecular responses of the HPA axis to acute social defeat stress (ASDS), we subjected mice to a single 1 h acute social defeat stress followed by euthanasia and tissue collection. RNA sequencing analysis was performed on the hypothalamus, pituitary gland, and adrenal gland from both the acute social defeat stress and control groups. Hypothalamic transcriptome profiling revealed a pronounced proinflammatory transcriptional signature after acute social defeat stress (Figure 2A). The volcano plot demonstrated significant up-regulation of 183 genes, enriched in proinflammatory cytokines (e.g., *Il-6*, *Cxcl10*), immunomodulatory molecules (*Icam1*, *Lcn2*), and immediate-early response genes (*Fosl2*, *Egr2*, *Egr3*, *Fosb*) in the acute social defeat stress group. A Kyoto Encyclopedia of Genes and Genomes (KEGG) pathway analysis revealed robust activation of the NF-κB signaling pathway, antigen processing and presentation, ECM-receptor interaction, and IL-17/TNF signaling pathways (Figure 2D). A Gene Ontology (GO) enrichment analysis revealed distinct pathway alterations in each HPA-axis organ under the acute social defeat stress. In the hypothalamus, up-regulated genes were enriched in pathways related to leukocyte cell–cell adhesion and regulation of T-cell activation, while down-regulated genes were associated with processes such as axon ensheathment, glial cell differentiation, and gliogenesis (Appendix A). In the pituitary gland, stress-induced transcriptional responses were marked by up-regulation of 990 stress-responsive genes (Figure 2B). Highlighted regulators included immediate-early response genes (*Egr2*, *Fosl1*, *Fosb*) and chemokines (*Ccl2*, *Cxcl2*, *Cxcl1*). A KEGG pathway analysis further revealed significant enrichment in the TNF signaling pathway and cytokine–cytokine receptor interaction (Figure 2E). Notably, pathways related to hormone synthesis, such as growth hormone synthesis and parathyroid hormone synthesis, also exhibited up-regulated trends. The GO enrichment analysis further indicated that up-regulated genes were enriched in metabolic processes, while pathways such as extracellular matrix organization and cellular component organization were significantly down-regulated (Appendix A).

The adrenal gland transcriptome analysis revealed distinct patterns with 1572 up-regulated genes, including markers of macrophage infiltration (*Ccl2*), angiogenic factors (*Vgf*), immediate-early response genes (*Fosl1*, *Egr3*, *Egr4*), and glucocorticoid regulators (*Il-6*, *Sst*) (Figure 2C). The KEGG pathway analysis confirmed activation of TNF signaling, NF-κB, and cytokine–cytokine receptor interaction pathways (consistent with hypothalamic and pituitary patterns), alongside adrenal-specific enrichment in growth hormone synthesis and thyroid hormone signaling pathways (Figure 2F). The GO analysis further showed that up-regulated genes were enriched in ameboidal-type cell migration and responded to peptide hormone, while Wnt signaling pathway-related processes were down-regulated (Appendix A).

Collectively, these findings demonstrate that acute social defeat stress induces transient inflammatory plasticity in the hypothalamus, accompanied by concurrent activation of both innate (TNF/IL-17 signaling pathways) and adaptive immune pathways. These stress-induced molecular changes may contribute to stress-related pathophysiological processes.

To identify differentially expressed genes (DEGs) involved in inflammatory pathways during acute social defeat stress, we performed a network visualization analysis of these pathways. This visualization clearly illustrated the key genes contributing to inflammatory responses across the HPA axis. In the hypothalamus, significant up-regulation was observed in *Lcn2* ($\log_{2}FC$ = 4.96), *Icam1* ($\log_{2}FC$ = 2.77), and *Ptgs2* ($\log_{2}FC$ = 2.36). In the pituitary gland, the most prominently up-regulated genes included *Cxcl1* ($\log_{2}FC$ = 6.57), *Cxcl2* ($\log_{2}FC$ = 5.88), *Ccl2* ($\log_{2}FC$ = 5.48), *Fosb* ($\log_{2}FC$ = 3.78), and *Fosl1* ($\log_{2}FC$ = 3.70). In the adrenal gland, notable increases were detected in *Il-6* ($\log_{2}FC$ = 5.69), *Fosl1* ($\log_{2}FC$ = 4.68), *Ccl2* ($\log_{2}FC$ = 4.31), and *Ccl7* ($\log_{2}FC$ = 4.25) (Figure 2G). Additional network analysis of hormone-related pathways in the pituitary and adrenal glands revealed mixed expression patterns in the corticosterone synthesis and secretion pathway: up-regulation of *Cyp11b1* and *Cyp21a1* was accompanied by down-regulation of *Nr4a1* and *Adcy4* (Figure 2H), suggesting potential pathway dysregulation.

The volcano plot analysis revealed up-regulation of immediate-early response genes, which was further confirmed across the HPA axis. Consistent up-regulation of *Fosl1*, *Fosb*, *Egr3*, and *Fosl2* was observed in the hypothalamus, pituitary gland, and adrenal gland (Figure 2I). This widespread transcriptional activation suggests a conserved, organ-wide mechanism for rapid stress-signaling transduction, potentially coordinating the HPA axis’s integrated response to acute social defeat stress. To validate these findings, we performed qPCR on selected immediate-early response genes (*Egr1*, *Fosl1*, and *Fosl2*) and inflammatory genes (*Cxcl1*, *Ccl2*, *Cxcl10*, *Il-6*, *Tnfrsf1b*, *Retnlg*, *S100a8*, and *S100a9*) across the hypothalamus, pituitary gland, and adrenal gland (Appendix A). The results of the qPCR confirmed up-regulation of these genes in acute social defeat-stressed mice, consistent with the RNA-seq results. These findings confirm the robust activation of immediate-early response genes and inflammatory mediators across the HPA axis under acute social defeat stress (Figure 2J).

### 2.3. Chronic Social Defeat Stress (CSDS)-Induced Molecular Alterations in the HPA Axis

To investigate the transcriptional landscape of chronic social defeat stress (CSDS) on the HPA axis, we performed RNA-seq analysis on the hypothalamus, pituitary gland, and adrenal gland of mice subjected to 10 days of CSDS. In the hypothalamus, 711 genes were down-regulated, including *Sstr4* ($\log_{2}FC$ = −2.76), *Chrna5* ($\log_{2}FC$ = −2.37), *Ghsr* ($\log_{2}FC$ = −1.08), and *Cckar* ($\log_{2}FC$ = −0.69). Conversely, 779 genes were up-regulated, including *Lcn2* ($\log_{2}FC$ = 5.73), *Cck* ($\log_{2}FC$ = 1.67), *Agrp* ($\log_{2}FC$ = 1.02), and *Npy* ($\log_{2}FC$ = 0.95) (Figure 3A). The KEGG pathway analysis revealed enrichment of down-regulated genes in neuroactive ligand–receptor interaction, cholinergic, GABAergic, and glutamatergic synapse pathways (Figure 3D). The GO analysis indicated that up-regulated genes were enriched in oxidative phosphorylation, extracellular matrix organization, and extracellular structure organization, while down-regulated genes were associated with neurotransmitter regulation and membrane potential (Appendix A).

In the pituitary gland, 862 genes were down-regulated, including *Cd19* ($\log_{2}FC$ = −2.27), *Ccl17* ($\log_{2}FC$ = −1.52), and *Stat4* ($\log_{2}FC$ = −1.15) (Figure 3B). These down-regulated genes were enriched in the glutamatergic and dopaminergic synapse pathways (Figure 3E). The GO analysis revealed that up-regulated genes were enriched in synapse structure and assembly (Appendix A). In the adrenal gland, 207 genes were down-regulated, including *Dr4* ($\log_{2}FC$ = −1.45) and *Cxcr4* ($\log_{2}FC$ = −1.26), while 717 genes were up-regulated, including *Gh* ($\log_{2}FC$ = 2.66), *Egr1* ($\log_{2}FC$ = 1.28), *Pdyn* ($\log_{2}FC$ = 2.53), and *S100a8* ($\log_{2}FC$ = 2.11) (Figure 3C). The KEGG analysis showed significant enrichment in the dopaminergic synapse, cholinergic synapse, and serotonergic synapse pathways (Figure 3F). The GO analysis further indicated that up-regulated genes were enriched in neurotransmitter transport and axonogenesis (Appendix A).

To identify genes involved in neurotransmitter-related pathways, we constructed a network visualization of DEGs. In the hypothalamus, genes in the neuroactive ligand–receptor interaction pathway (KEGG) were significantly down-regulated, with a notable subset of ligand–receptor pairs showing dysregulation (Figure 3G). Further analysis of neurotransmitter ligand–receptor pairs revealed the following: glutamate-related genes: partial up-regulation of ligands/receptors; GABA-related genes: down-regulation of both ligands and receptors; choline acetyltransferase (ChAT)-related genes: down-regulation of both ligand synthesis enzymes and receptors. In contrast, dopaminergic and serotonergic synapse-related genes were not significantly dysregulated in the hypothalamus (Figure 3H).

In the pituitary gland, down-regulated genes were enriched in the glutamatergic and dopaminergic synapse pathways with no significant dysregulation of other neurotransmitter systems. In the adrenal gland, up-regulated genes were enriched in the neuroactive ligand–receptorr interaction pathway, particularly showing significant up-regulation of dopamine and norepinephrine receptor-related genes (Figure 3G,H).

To explore hormone-related dysregulation, we analyzed ligand–receptorr pairs in DEGs (Figure 3I). In the hypothalamus, growth hormone-releasing hormone (*Ghrh*) was up-regulated, but its receptors (*Ghr*, growth hormone receptor; *Ghsr*, growth hormone secretagogue receptor) were down-regulated. Cholecystokinin (*Cck*) was up-regulated, but its receptors (*Cckar*, CCK-A receptor; *Cckbr*, CCK-B receptor) were down-regulated. Arginine vasopressin (*Avp*) was up-regulated, but its receptor *Avpr1a* (arginine vasopressin receptor 1A) was down-regulated. Neuropeptide Y (*Npy*) and its receptor *Npy5r* were up-regulated, while other receptors (*Npy1r*, *Npy2r*) were down-regulated. Agouti-related protein (*Agrp*) was up-regulated, but its receptors *Mc3r* and *Mc4r* (melanocortin receptors) were down-regulated. In the pituitary and adrenal glands, receptors for *Agrp* (*Mc3r*, *Mc4r*), *Npy* (*Npy5r*), and growth hormone (*Ghr*) were up-regulated, suggesting organ-specific compensatory mechanisms (Figure 3J).

### 2.4. Comparative Analysis of Acute and Chronic Social Defeat Stress Responses in the HPA Axis

Through RNA-seq analysis combined with existing evidence, we observed distinct transcriptional profiles between acute and chronic social defeat stressors on the HPA axis. In acute social defeat stress, up-regulated genes across the HPA axis were enriched in inflammatory pathways (Figure 4A). However, these genes showed no consistent expression trends in the hypothalamus under chronic social defeat stress (Figure 4B,C). Notably, neurotransmitter-related pathways exhibited contrasting patterns: chronic social defeat stress down-regulated cholinergic synapse and GABAergic synapse genes in the hypothalamus and pituitary gland, while up-regulating these pathways in the adrenal gland alongside dopaminergic and serotonergic synapse enrichment (Figure 4D). In acute social defeat stress, partial up-regulation of cholinergic and GABAergic synapse genes was observed in the hypothalamus, suggesting transient synaptic activity (Figure 4E,F).

Additionally, hormone synthesis pathways differed between stress durations: acute social defeat stress enriched growth hormone synthesis (pituitary) and parathyroid hormone synthesis (pituitary), whereas chronic social defeat stress only enriched adrenal pathways like corticosterone synthesis and secretion and aldosterone synthesis and secretion (Figure 4G). Gene expression profiles further revealed that growth hormone-related genes were consistently up-regulated in acute social defeat stress but showed no coordinated changes in chronic social defeat stress, highlighting pathway dysregulation under prolonged stress (Figure 4H,I). These findings underscore organ-specific and duration-dependent transcriptional adaptations in HPA-axis responses to stress.

Through integration of the CellChat and CellPhoneDB databases, we systematically analyzed ligand–receptor interactions involving hormones, neurotransmitters, and neuropeptides in chronic and acute social defeat stress. In chronic social defeat stress, the majority of ligands (e.g., *Gh*, *Npy*, *Agrp*) exhibited up-regulated expression, with the exception of *Ghrl* (ghrelin) and *Gal* (galanin), which were down-regulated (Figure 4J). In contrast, acute social defeat stress induced opposite directional changes in ligand–receptor pairs compared to chronic social defeat stress (e.g., *Ghrl* up-regulation and *Npy* down-regulation in acute social defeat stress versus *Ghrl* down-regulation and *Npy* up-regulation in chronic social defeat stress). This suggests that chronic social defeat stress drives a broader and more persistent rewiring of ligand–receptor communication networks compared to acute social defeat stress, particularly in hormone and neuropeptide signaling pathways (Appendix A).

## 3. Discussion

### 3.1. Distinct Activation Patterns of the HPA Axis Under Acute vs. Chronic Social Defeat Stress

The present study investigated the distinct activation patterns of the HPA axis under acute and chronic social defeat stress conditions and their physiological consequences. We observed differential HPA-axis responses to these stress conditions, resulting in divergent outcomes in inflammation, neuronal activity, and hormone regulation. These findings provide novel insights into the neuroimmune mechanisms underlying stress-related mental health disorders, such as anxiety and depression.

Under acute social defeat stress, mice exhibited transient anxiety- and depression-like behaviors that resolved after a short recovery period. RNA-seq analysis revealed increased systemic inflammation, marked by up-regulation of pro-inflammatory cytokines like *Cxcl10* and *Il-6*, consistent with previous findings that acute social defeat stress triggers a robust neuroimmune response mediated by the HPA axis [3,43]. Additionally, rapid activation of immediate-early genes (IEGs), such as *Egr1* and *Fosl2*, was observed; these are known to regulate neuronal activity and inflammatory responses under stress conditions, suggesting that acute social defeat stress induces a transient but robust inflammatory response that contributes to short-term behavioral and physiological changes [44,45].

In contrast, chronic social defeat stress led to persistent anxiety- and depression-like behaviors, accompanied by long-term alterations in neuroimmune regulation. RNA-seq analysis revealed significant enrichment of the interleukin (IL)-17 signaling pathway, with pronounced up-regulation of the *Lcn2* gene. Previous studies have shown that *Lcn2* links peripheral inflammation to central nervous system (CNS) dysfunction, particularly in anxiety disorders [46]. The sustained activation of the IL-17 pathway in chronic social defeat stress conditions suggests a maladaptive neuroimmune response, which may underlie the persistent behavioral changes observed [47,48]. These findings highlight the importance of investigating the IL-17 pathway and associated genes in understanding the bidirectional communication between the periphery and CNS during chronic social defeat stress.

The distinct activation patterns of the HPA axis under acute and chronic social defeat stress have significant implications for stress-related mental health disorders. The findings underscore the need to consider stress duration in evaluating neuroimmune responses and their effects on mental health. Furthermore, identifying Lcn2 and the IL-17 pathway as key regulators of chronic social defeat stress-induced inflammation provides potential therapeutic targets for addressing neuroimmune dysregulation and alleviating the long-term consequences of chronic social defeat stress. These insights deepen our understanding of stress-related pathophysiology and pave the way for personalized treatments and early interventions for stress-associated disorders.

### 3.2. Chronic Social Defeat Stress-Induced Neurotransmitter Imbalance

We also observed that chronic social defeat stress induces significantly different patterns of neurotransmitter changes compared to acute social defeat stress, and these changes were not observed in acute social defeat stress paradigms. Specifically, in the hypothalamus, levels of γ-aminobutyric acid (GABA) decreased significantly, while levels of glutamate increased, and the activity of choline acetyltransferase (ChAT) also altered. These changes reflect the profound impact of chronic social defeat stress on neurochemical homeostasis and provide critical insights into the mechanisms linking chronic social defeat stress to anxiety and depression.

GABA, the primary inhibitory neurotransmitter, plays a crucial role in maintaining the excitatory/inhibitory balance necessary for normal brain function [49]. The observed reduction in GABA levels during chronic social defeat stress disrupts this balance and promotes neuronal hyperexcitability, which is linked to anxiety-like behaviors through heightened emotional reactivity and impaired emotional regulation [50]. Furthermore, this effect is exacerbated by the increase in glutamate, the primary excitatory neurotransmitter, which regulates synaptic plasticity and information processing. Excessive glutamate release can induce neurotoxic effects, including excessive activation of NMDA receptors and oxidative stress, contributing to the development of depression-like behaviors [51,52]. These findings align with clinical observations that patients with major depressive disorder (MDD) exhibit similar glutamate and GABA abnormalities, suggesting shared neurochemical mechanisms between chronic social defeat stress and mood disorders [53].

Furthermore, the altered activity of ChAT, which synthesizes acetylcholine (ACh), highlights broader neurotransmitter dysregulation during chronic social defeat stress. ACh, a key modulator of cognitive and emotional functions, is implicated in stress-induced neuroplasticity and emotional regulation [54]. Changes in ChAT activity suggest that chronic social defeat stress may disrupt cholinergic signaling, potentially contributing to cognitive deficits and emotional dysregulation observed in chronic social defeat stress models.

Our findings reveal that chronic social defeat stress profoundly impacts the central nervous system by disrupting the balance between inhibitory and excitatory neurotransmitters. The reduction in GABA, the increase in glutamate, and changes in cholinergic activity collectively impair neuronal function and networks closely linked to the emergence of anxiety and depression. These findings underscore the critical need for research into the molecular mechanisms driving neurotransmitter dysregulation during chronic social defeat stress and the development of therapeutic strategies to restore neurochemical balance and mitigate stress-related neuropsychiatric disorders.

### 3.3. Dynamic Regulation of HPA-Axis Hormone Synthesis and Secretion Under Acute and Chronic Social Defeat Stress

Moreover, this study elucidates the dynamic regulatory mechanisms of the HPA axis in response to acute and chronic social defeat stress, focusing on hormone synthesis and secretion patterns. Under acute social defeat stress, both the sympathetic nervous system (SNS) and HPA axis are activated, leading to increased secretion of growth hormone (GH) and corticosterone. This response supports the “fight-or-flight” reaction by fulfilling energy demands and modulating immune responses [17]. Our findings reveal significant enrichment of GH synthesis pathways in the pituitary and adrenal glands during acute social defeat stress. The transcriptional activation of *Fos*, *Junb*, and *Adcy4* facilitates GH transcription, while *Socs1* mild up-regulation prevents GH overproduction [55]. Additionally, *Sstr2* enhances sensitivity to somatostatin (SST), ensuring precise regulation of GH release. Simultaneously, both corticosterone-related pathways are enriched in the pituitary and adrenal glands. In the adrenal gland, increased transcription levels of *Nr5a1*, *Cyp11b2*, and *Ldlr*, along with decreased levels of *Mc2r*, *Nr0b1*, and *Plcb1*, suggest that acute social defeat stress triggers synchronized activation of key synthetic enzymes and substrate uptake systems, enabling burst corticosterone secretion. This process is fine-tuned by receptor sensitivity and signaling pathways, balancing rapid corticosterone synthesis with negative feedback mechanisms to prevent excessive tissue damage [56].

In contrast, chronic social defeat stress disrupts the body’s compensatory mechanisms, transitioning to systemic suppression of HPA-axis activity. Our results reveal that chronic social defeat stress suppresses key hormone pathways, including GH, corticosterone, thyroid hormones, and aldosterone. This suppression is mediated by reduced transcription levels of *Ghrl* and *Ghsr* in the hypothalamus [57]. Diminished hypothalamic *Ghrl/Ghsr* signaling directly inhibits GHRH neuron activity, reducing GH synthesis in the pituitary gland. Moreover, *Ghrl’s* dual role in regulating appetite and energy homeostasis may indirectly influence GH synthesis by reducing feeding behaviors, further exacerbating the effect of chronic social defeat stress [58]. Notably, in chronic social defeat stress conditions, reduced expression of *Scarb1*, *Star*, *Cyp11a1*, *Hsd3b1/Hsd3b6*, *Hsd11b1*, and *Srd5a2* in the adrenal gland suggests that chronic social defeat stress disrupts steroidogenesis through mechanisms that impair cholesterol uptake, pregnenolone generation, progesterone conversion, and corticosterone activation, resulting in cascading suppression of corticosterone synthesis and secretion [59]. These findings highlight the complexity of HPA-axis regulation under stress, emphasizing the role of transcriptional networks and hormonal feedback mechanisms in stress adaptation and pathological states.

### 3.4. Clinical Implications

The findings of the present study provide critical insights into the neuroimmune and neurochemical mechanisms underlying chronic stress responses. The sustained up-regulation of the IL-17 pathway and Lcn2 in chronic stress mirrors clinical observations in patients with anxiety or depression [60,61,62]. IL-17A has been identified as a potential biomarker for forecasting responses to specific antidepressants [63]. A targeted anti-IL-17A approach using Ixekizumab has shown promise in reducing depressive symptoms in 40% of psoriasis patients with co-occurring MDD [64]. Conversely, blocking IL-17A signaling via its receptor (e.g., through Brodalumab therapy) has been linked to adverse psychiatric outcomes, including increased suicide risk, in psoriasis patients [65]. Preclinical studies in rodents corroborate these findings, showing that IL-17A administration induces depressive-like behaviors [48] and stress exposure elevates IL-17A levels [66,67,68,69]. These suggest that targeting this marker might provide antidepressant actions. Inflammation is likely a key disease modifier that increases vulnerability to stress. Modulating inflammation may offer a holistic therapeutic benefit, applicable not only to acute stress responses but also to scenarios involving a combination of multiple factors.

Chronic stress-induced imbalances in GABA and glutamate, as observed in our study, align with findings from human neuroimaging and post-mortem analyses, which consistently show decreased GABAergic signaling and heightened glutamatergic activity within the prefrontal cortex and limbic structures of individuals with major depressive disorder (MDD) [52,53]. This consistent evidence supports the idea that the molecular markers identified in our research could serve as potential diagnostic and therapeutic targets for addressing stress-related disorders.

In conclusion, identifying robust diagnostic and treatment response biomarkers in stress-related disorders (e.g., MDD) remains challenging due to the disorders’ clinical diversity and the broad variability in treatment outcomes. Future research should focus on developing combinatorial approaches to identify robust biomarkers and refine diagnostic and therapeutic strategies for personalized management of these conditions.

## 4. Materials and Methods

### 4.1. Animals

Male C57BL/6 mice (8–10 weeks old, 22–24 g) and ICR mice (25 weeks old, 40–45 g) were purchased from the Huazhong Agricultural University Experimental Animal Center. The mice were housed under temperature-controlled conditions (22 ± 2 °C) and humidity-controlled conditions (50 ± 5%) with a 12 h light/dark cycle. C57BL/6 mice were housed in groups (4 to 5 per cage), unless they were undergoing acute or chronic social defeat stress protocols. ICR mice were individually housed (1 mouse/cage) to preserve their natural aggression and avoid the formation of a social hierarchy that could reduce the frequency of attacks. The animals were sacrificed humanely at the end of the experiments to minimize suffering.

### 4.2. Social Defeat Stress

Several animal models have been developed to investigate the effects of stress on the physiology, immunity, and behavior of rodents. Among these are the chronic social defeat stress (CSDS) [4,5], chronic unpredictable mild stress (CUMS) [6,7,8,9], chronic restraint stress (CRS) [10,11,12,13,14], and the learned helplessness models [15,16]. The CSDS model effectively simulates social stress-induced anxiety and depression-like behaviors. Traditionally, male mice have been the preferred subjects in stress research due to their higher model reliability. However, recent studies have introduced modified protocols for female mice, with a novel chronic social stress paradigm in female mice integrating multiple social-stress components, including resident–intruder interactions, social crowding, and social instability [70]. In addition, the social variable chronic stress (SVCS) model has been shown to be more suitable for detecting sex-specific responses to chronic stress, especially when comparing behavioral and neurobiological outcomes between males and females. Each model offers unique advantages for studying different aspects of stress biology.

In the present study, both acute and chronic social defeat stress paradigms were employed to explore stress effects on the HPA axis and behavior. To investigate the behavioral and transcriptional responses of the HPA axis to milder social defeat stress, we adopted both the subthreshold stress model and the initial stress-stage model as the foundation for our acute social defeat stress (ASDS) model [5,38]. Consistently, both models utilized male mice as subjects. To maintain model reproducibility and comparability, we selected the acute social defeat stress (ASDS) model and the classical chronic social defeat stress (CSDS) model for our study, both employing a male-mouse social stress paradigm.

#### 4.2.1. Acute Social Defeat Stress (ASDS)

Acute social defeat stress (1 h): C57BL/6 mice were placed in the cage of a more aggressive ICR mouse, allowing direct physical contact for 5 min. Subsequently, the mice were separated by a perforated acrylic board for 10 min, providing a psychological stress cue. This 15 min cycle was repeated four times within 1 h. After the acute social defeat stress session, C57BL/6 mice were removed for behavioral assessments and tissue sample collection.

Acute social defeat stress (24 h): The procedure was identical to the 1 h acute social defeat stress model. After completing the four stress cycles, C57BL/6 mice were returned to their home cages for a 23 h recovery period before undergoing behavioral assessments and tissue sample collection.

#### 4.2.2. Chronic Social Defeat Stress (CSDS)

For the chronic social defeat stress (CSDS) models, we strictly followed the experimental protocol established by Krishnan et al. and the well-characterized procedure [4,5]. In short, the chronic social defeat stress lasted for 10 days. The C57BL/6 mice and ICR mice were adaptively housed for 3 days prior to the experiment. Among the ICR mice, those with the highest aggression levels were selected for model construction. Each day, a C57BL/6 mouse was placed into the cage of an ICR mouse. Due to territorial instincts, the ICR mouse exhibited directed aggression toward the C57BL/6 mouse. After 5 min of direct contact, a perforated acrylic board was used to separate the two mice, allowing for 24 h of psychological stress. This procedure was repeated daily for 10 consecutive days, with a different unexposed ICR mouse used each day. The number of attacks on each C57BL/6 mouse was recorded daily, ensuring that each C57BL/6 mouse experienced a similar frequency of attacks. After 10 days, the C57BL/6 mouse was returned to its original cage for a 24 h recovery period before undergoing behavioral assessments and tissue sample collection.

### 4.3. Behavioral Studies

All behavioral tests were conducted in a testing room maintained at constant temperature, equipped with appropriate lighting, and designed to minimize noise and stress. Mice were placed in their home cages 1 h prior to testing to acclimate the environment. All behavioral assessments were performed by experimenters blinded to the group assignments. The testing order was randomized, and all experimental apparatus was cleaned with 75% alcohol after each session to eliminate potential olfactory cues. Mouse movements were monitored using the PanLab SMART video tracking system (PanLab/Harvard Apparatus, Barcelona, Spain).

### 4.4. Social Interaction Test (SIT)

The social interaction test (SIT) was conducted in a white test box (42 cm × 42 cm × 42 cm) with a defined social zone (14 cm × 24 cm rectangular area along the top center of the box) for housing an ICR mouse and two corner zones (9 cm × 9 cm squares) at the lower left and lower right corners according to previous research [39]. All behavioral assessments were performed by experimenters blinded to the group assignments. The experiment consisted of two phases. In the first phase, a transparent perforated box was placed in the social zone, and the time spent by the C57BL/6 mouse in the social zone over 2 min and 30 s was recorded. In the second phase, an unfamiliar ICR mouse was placed in the perforated box, and the same procedures were followed. The social interaction index (SIT ratio) was calculated as the ratio of time spent in the social zone during the second phase to the first phase. Mouse trajectories were tracked using the SMART video system (PanLab/Harvard Apparatus, Barcelona, Spain).

### 4.5. Open Field Test (OFT)

The open field test (OFT) utilized an automated tracking system (PanLab/Harvard Apparatus, Barcelona, Spain) to assess locomotor activity and anxiety-related behaviors in C57BL/6 mice within a 40 cm × 40 cm × 40 cm open field arena. The arena was divided into a central 20 cm × 20 cm “anxiety-sensitive” zone and four peripheral 10 cm × 10 cm corner regions according to previous research [40]. All behavioral assessments were performed by experimenters blinded to the group assignments. Mice were gently placed at the center boundary, oriented toward a fixed wall, and allowed to explore for 10 min following a 5 min acclimation period. The automated analysis system quantified total distance traveled (cm), number of entries into the central zone, time spent in the central zone (s), and time spent in the central and peripheral zones (s). Standardized experimental conditions ensured consistency in animal positioning, arena dimensions, and data collection.

### 4.6. Adrenalectomy (ADX)

Adrenalectomy was performed on experimental C57BL/6 mice (ADX group) according to previous research [41]. Under deep anesthesia, a small abdominal incision was made, and the left and right adrenal glands were carefully removed to avoid blood vessels and minimize tissue damage. After removal, the abdominal incision was sutured, and the mice were placed back in their original cages to recover under appropriate temperature conditions, with supplementary hydration and nutrition provided. C57BL/6 mice served as the sham group and underwent a sham surgery, which involved making a small incision in the abdominal cavity followed by suturing, without further intervention. Both groups of mice were given 4 days to recover before undergoing behavioral assessments and tissue sample collection.

For the priming treatment following adrenalectomy, both the sham group and ADX-model group mice were placed into the ICR cage for 5 min of direct contact. Afterward, the two mice were separated by a perforated acrylic board, and psychological stress was administered for 55 min. After priming treatment, the C57BL/6 mice were returned to their home cages for a 12 h recovery period before undergoing behavioral assessments and tissue sample collection.

### 4.7. Serum Corticosterone Enzyme-Linked Immunosorbent Assay (ELISA)

Serum corticosterone levels were measured using an ELISA kit (D-22335 Hamburg, Germany). Mice from the control (untreated), sham- (sham surgery), and adrenalectomy-model groups were anesthetized, and whole blood was collected from the eyeball. The blood was allowed to separate at room temperature for 30 min. Following centrifugation, the supernatant was collected, and the serum was stored at −80 °C. Corticosterone levels were measured using the (TECAN, RE52211, Hamburg, Germany) kit, with subsequent procedures carried out according to the protocol provided by the kit.

### 4.8. Flow Cytometry

Single-cell suspensions from mice spleen, peripheral blood, and bone marrow were prepared following tissue-specific digestion protocols.
Spleen: Mechanically homogenized and filtered through a 70 µm nylon mesh (Becton Dickinson, Franklin Lakes, NJ, USA) to isolate cellular components.Blood: Anticoagulated peripheral blood collected via retro-orbital puncture and processed immediately.Bone marrow: Flushed with 2% BSA-PBS to release hematopoietic progenitor cells.
Antibody panel:
Anti-mouse CD45 (Cat. #1031557, BioLegend, San Diego, CA, USA);Anti-mouse CD11b (Cat. #101207, BioLegend, San Diego, CA, USA);Anti-mouse CD19 (Cat. #152409, BioLegend, San Diego, CA, USA);Anti-mouse CD3 (Cat. #100213, BioLegend, San Diego, CA, USA);Anti-mouse CD4 (Cat. #100408, BioLegend, San Diego, CA, USA);Anti-mouse CD8 (Cat. #100803, BioLegend, San Diego, CA, USA);Anti-mouse Ly6G (Cat. #127613, BioLegend, San Diego, CA, USA);Anti-mouse Ly6C (Cat. #128005, BioLegend, San Diego, CA, USA).
Protocol:
Stained with antibodies (1:100–1:50 dilution) for 30 min at 4 °C.Washed three times in PBS with 0.1% formaldehyde.Analyzed using Beckman Cytoflex S with CytExpert software 2.6.0.105 (Beckman Coulter, Inc., Brea, CA, USA).
Gating strategy: Defined by forward/side scatter (Appendix A).

### 4.9. Tissue Collection

Mice were euthanized 24 h after the final acute social defeat stress exposure or at the conclusion of a 10-day social stress protocol. Subsequently, the hypothalamus, pituitary gland, and adrenal gland were excised under aseptic conditions, immediately snap-frozen in liquid nitrogen, and stored at −80 °C until RNA isolation. To ensure biological reproducibility, two independent experimental replicates were included per group. Total RNA was isolated using a total RNA isolation reagent (TRIzol, Thermo Fisher Scientific, Waltham, MA, USA) and RNA integrity was assessed using the Agilent Bioanalyzer system (Agilent Technologies, Santa Clara, CA, USA) (RNA integrity number ≥ 8.0).

### 4.10. Quantitative Real-Time PCR (qRT-PCR)

To confirm differentially expressed genes (DEGs) in the hypothalamic, pituitary gland, and adrenal gland, a qRT-PCR-based verification strategy was employed. RNA samples were converted into complementary DNA (cDNA) using the ABScript II RT Mix with gDNA Remover Kit (RK20403, ABclonal, Wuhan, China) following established protocols. The qRT-PCR reactions were conducted on the Applied Biosystems ABI 7300 Real-Time PCR System (Applied Biosystems, Foster City, CA, USA) utilizing the SYBR Green PCR Master Mix (RK21203, ABclonal, China). Gene expression levels were normalized to the housekeeping gene GAPDH, with fold changes calculated using the $2^{-\Delta \Delta C_{t}}$ method. All experiments were independently replicated three times to ensure experimental reproducibility.

### 4.11. RNA-Seq Library Preparation and Sequencing

mRNA capture beads (VAHTS mRNA-seq v2 Library Prep Kit for Illumina, Vazyme, Nanjing, China) were used to extract mRNA from total RNA. Fast RNA-seq Lib Prep Kit V2 (Cat. No. RK20306, ABclonal, Wuhan, China) was used to prepare cDNA libraries for Illumina sequencing according to the manufacturer’s protocol. Sequencing was performed using the 150 bp paired-end configuration on an Illumina NovaSeq X plus platform (Illumina, San Diego, CA, USA). The quality of the raw paired-end reads from RNA-seq were evaluated with FASTQC (https://www.bioinformatics.babraham.ac.uk/projects/fastqc/, accessed on 8 May 2025). The raw reads were processed through quality filtering and mapped to the mouse reference genome (mm39) using HISAT2 2.2.1 (alignment efficiency > 95%) [71]. The transcripts were quantified and annotated using the gencode.vM27.annotation.gtf file via HTseq-count 2.0.3 [72].

### 4.12. RNA-Seq Data Analysis

Differential gene expression analysis was conducted using DESeq2 with the following criteria: an absolute fold change ≥ 1.5 and *p*-value < 0.05 [73]. Enrichment analyses, including Gene Ontology (GO) and Kyoto Encyclopedia of Genes and Genomes (KEGG), were performed on the differentially expressed genes using the clusterProfiler 4.8.1 [74]. Pathways were considered significantly enriched if the false discovery rate (FDR) < 0.05. Network visualizations were generated using Cytoscape 3.10.3 [75]. Receptor–ligand interactions were analyzed using the CellChat 2.0.0 and CellPhoneDB V2 databases [76,77].

### 4.13. Data Availability

The raw sequencing data generated in this study have been submitted to the Genome Sequence Archive (GSA) hosted by the National Genomics Data Center (NGDC), Chinese Academy of Sciences, under accession number CRA025618. The dataset is publicly available through the following link: https://ngdc.cncb.ac.cn/gsa (accessed on 8 May 2025).

### 4.14. Statistical Analysis

Statistical analyses were performed using GraphPad Prism (v9.0) and R (v4.3.0). Data are presented as means ± standard error of the mean (SEM). Sample sizes are indicated in figure legends. For comparisons between two groups, unpaired *t*-tests or Mann–Whitney U tests were used. When comparing three or more groups, one-way ANOVA followed by Tukey’s post hoc test was used. *p*-values and significance thresholds (*p* < 0.05) are specified in the corresponding figure panels.

## 5. Conclusions

This study provides comprehensive insights into the dynamic regulation of the HPA axis under acute and chronic social defeat stress conditions, revealing distinct molecular mechanisms and neuroendocrine outcomes. The findings demonstrate that acute social defeat stress elicits a rapid and transient neuroendocrine response, characterized by increased hormone secretion, inflammation, and IEG activation, supporting the organism’s “fight-or-flight” mechanism. In contrast, chronic social defeat stress leads to long-term neuroendocrine dysregulation, including suppressed hormone synthesis, sustained neuroinflammation, and disrupted neurotransmitter balance, contributing to chronic behavioral and physiological alterations. The research highlights the critical role of the HPA axis in stress adaptation and mental health, providing novel therapeutic targets, such as the IL-17 pathway and neurotransmitter regulation, for addressing chronic social defeat stress-related disorders. These findings underscore the need for personalized interventions targeting specific stress pathways to restore neuroendocrine balance and improve mental health outcomes.

## Figures and Tables

**Figure 1 ijms-26-06063-f001:**
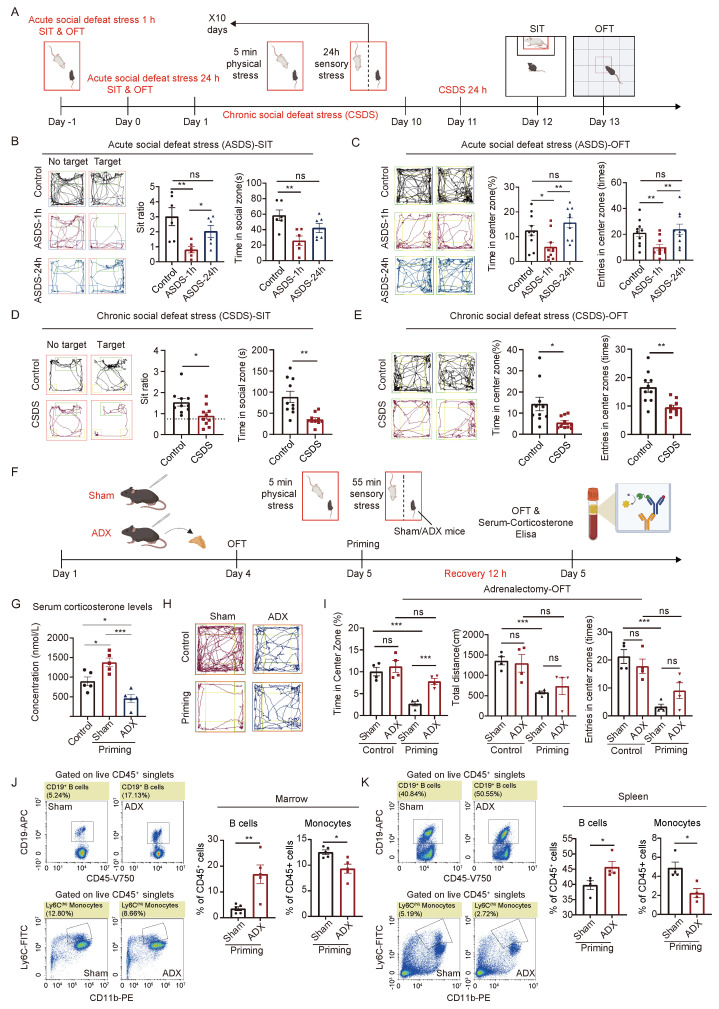
The HPA axis plays a crucial role in the body’s response to acute (ASDS) and chronic social defeat stress (CSDS). (**A**) Experimental procedures for constructing ASDS and CSDS models, along with methods for assessing anxiety- and depression-like phenotypes. (**B**) Social interaction test (SIT) path diagrams and statistical analysis of SIT ratios and time spent in the social zone under ASDS for 1 h and 24 h (control, *n* = 6; ASDS-1 h, *n* = 6; ASDS-24 h, *n* = 6). (**C**) Open field test (OFT) results, showing time spent in the central zone and entries into the central zone under ASDS for 1 h and 24 h (control, *n* = 10; ASDS-1 h, *n* = 10; ASDS-24 h, *n* = 10). (**D**) Social interaction test (SIT) path diagrams and statistical analysis of SIT ratios and time spent in the social zone under CSDS (control, *n* = 10; CSDS, *n* = 10). (**E**) Open field test (OFT) results, showing time spent in the central zone and entries into the central zone under CSDS (control, *n* = 10; CSDS, *n* = 10). (**F**) Experimental workflow for ADX, OFT, and serum corticosterone measurement. (**G**) Quantitative analysis of serum corticosterone. levels post-ADX and stimulation (control, *n* = 5; sham priming, *n* = 5; ADX priming, *n* = 5).
(**H**) Representative OFT trajectories of sham- and adrenalectomy-model mice pre- and post-priming.
(**I**) Statistical charts illustrating time spent in the central zone, total distance traveled, and entries
into the central zone during OFT testing pre- and post-priming for sham- and adrenalectomy-model
groups (sham priming, *n* = 4; ADX priming, *n* = 4). (**J**) Representative flow cytometry plots depicting CD19^+^ B lymphocytes and Ly6C^+^ monocytes in the bone marrow of sham and ADX mice
post-priming, along with corresponding statistical charts (sham priming, *n* = 5; ADX priming, *n* = 5).
(**K**) Representative flow cytometry plots of CD19^+^ B lymphocytes and Ly6C^+^ monocytes in the
spleen of sham and ADX mice following priming, along with statistical charts (sham priming, *n* = 4;
ADX priming, *n* = 4). All data are presented as mean ± SEM, analyzed by two-tailed Student’s t
test (**D**,**E**,**J**,**K**) and one-way ANOVA with Tukey’s post hoc test (**B**,**C**,**G**,**I**). Significance levels are
indicated as follows: ns, not significant; *, *p* < 0.05; **, *p* < 0.01; ***, *p* < 0.001; ns, not significant versus
untrained mice.

**Figure 2 ijms-26-06063-f002:**
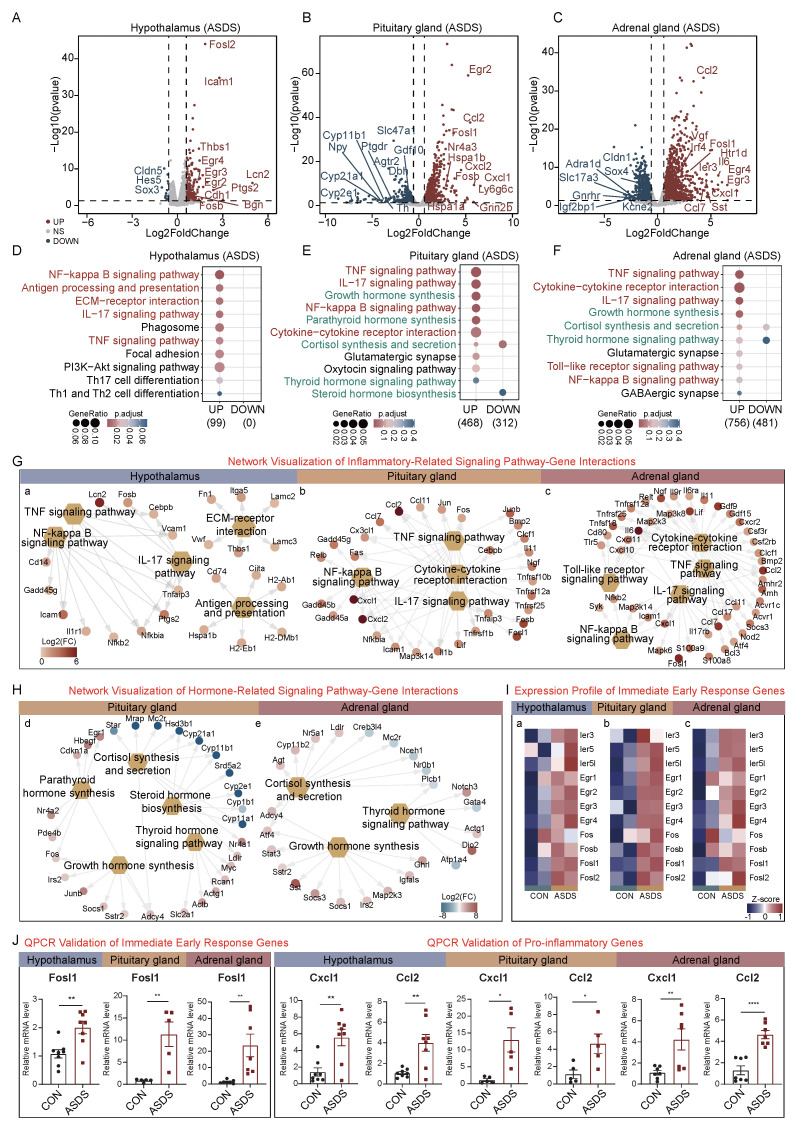
Acute social defeat stress (ASDS)-induced molecular alterations in the HPA axis. (**A**–**C**) Volcano plots showing DEGs in the hypothalamus, pituitary gland, and adrenal gland under ASDS (red: up-regulated genes; blue: down-regulated genes; gray: non-significant changes; fold change ≥ 1.5, *p* value < 0.05; The dotted lines represent the thresholds for significant up-regulation and down-regulation.). (**D**–**F**) KEGG analysis of DEGs in the hypothalamus, pituitary gland, and adrenal gland under ASDS. The text colors indicate specific categories of signaling pathways: Red represents inflammatory-related signaling pathways, Green represents hormone-related signaling pathways. (**G**) Network visualization of inflammatory pathway–gene interactions via KEGG analysis in the hypothalamus (**a**), pituitary gland (**b**), and adrenal gland (**c**) under ASDS. (**H**) Network visualization of hormone pathway–gene interactions via KEGG analysis in the pituitary gland (**d**) and adrenal gland (**e**) under ASDS. (**I**) Expression profile of immediate-early response genes in the hypothalamus, pituitary gland, and adrenal gland under ASDS. (**J**) Validation of immediate-early response and
pro-inflammatory gene expression in the hypothalamus, pituitary gland, and adrenal gland under
ASDS via qPCR. Significance levels are indicated as follows: ns, not significant; * *p* < 0.05; ** *p* < 0.01; **** *p* < 0.0001.

**Figure 3 ijms-26-06063-f003:**
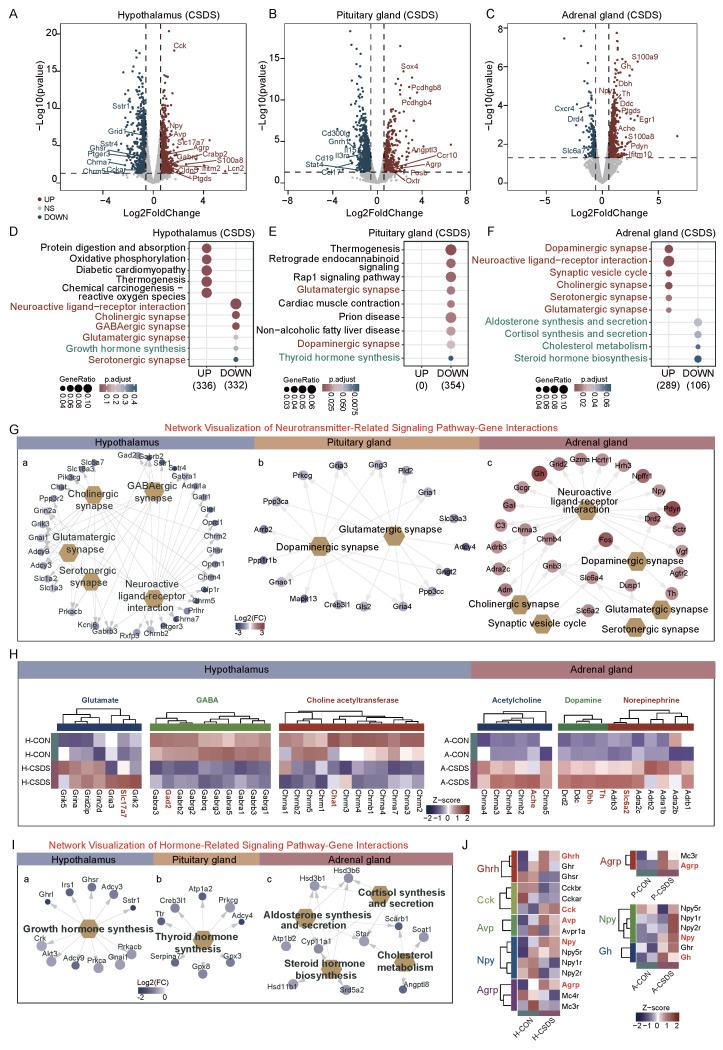
Molecular alterations in the HPA axis induced by chronic social defeat stress (CSDS). (**A**–**C**) Volcano plots illustrating DEGs in the hypothalamus, pituitary gland, and adrenal gland under CSDS (red: up-regulated genes; blue: down-regulated genes; gray: non-significant changes; fold change ≥ 1.5, *p* value < 0.05; The dotted lines represent the thresholds for significant up-regulation and down-regulation.). (**D**–**F**) KEGG pathway analyses of DEGs in the hypothalamus, pituitary gland, and adrenal gland under CSDS. The text colors indicate specific categories of signaling pathways: Red represents neurotransmitter-related signaling pathways, Green represents hormone-related signaling pathways. (**G**) Network visualizations showing interactions between synapse pathways and genes in the hypothalamus (**a**), pituitary gland (**b**), and adrenal gland (**c**), as identified by KEGG analysis. (**H**) Heatmaps display expression levels of genes associated with glutamate, GABA, choline acetyltransferase, acetylcholine, dopamine, and norepinephrine pathways in the hypothalamus (**left**) and adrenal gland (**right**). The color scale indicates the Z-score, ranging from −2 to 2. (**I**) Network
visualizations illustrating interactions between hormone pathways and genes in the hypothalamus
(**a**), pituitary gland (**b**), and adrenal gland (**c**), as identified by KEGG analysis. (**J**) Heatmaps illustrate expression levels of genes related to ghrelin (*Ghrl*), agouti-related protein (*Agrp*), cholecystokinin (*Cck*),
avasopressin (*Avp*), neuropeptide Y (*Npy*), and growth hormone (*Gh*) pathways in the hypothalamus (**left**), pituitary gland (**upper right**), and adrenal gland (**lower right**). The color scale indicates the Z-score, ranging from −2 to 2.

**Figure 4 ijms-26-06063-f004:**
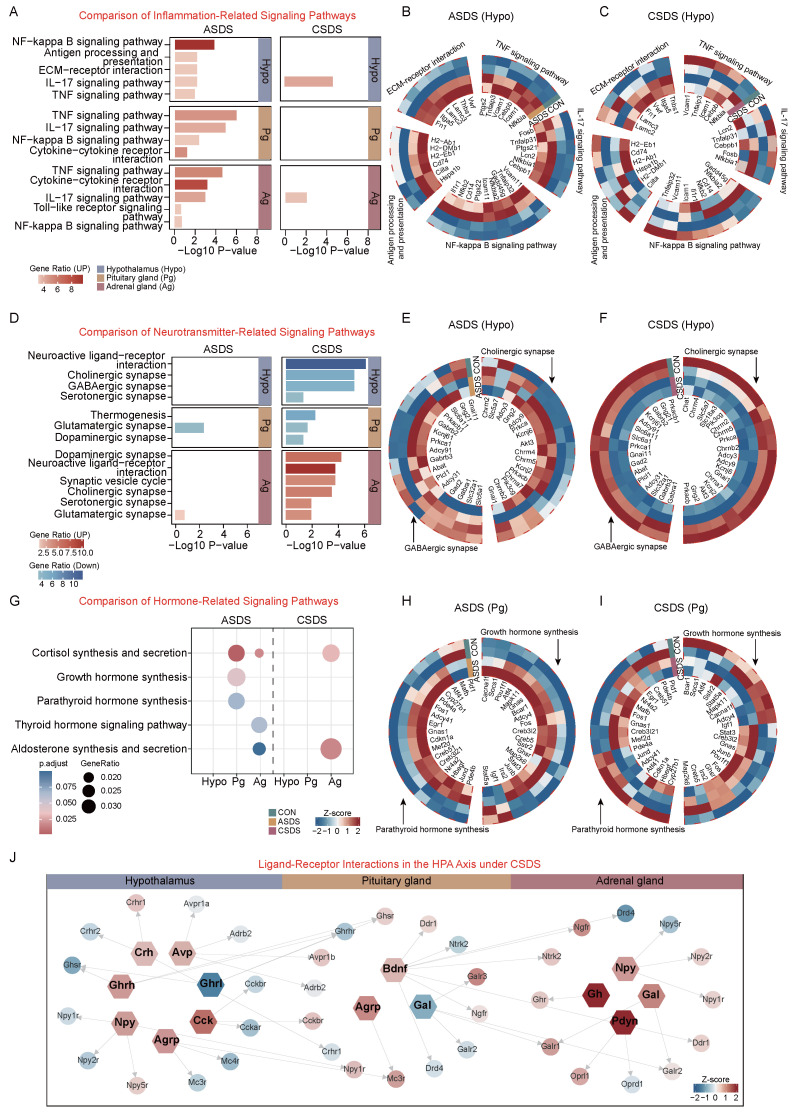
Comparative analysis of stress-induced pathway dysregulation in the HPA axis. (**A**) Comparative analysis of inflammation-related signaling pathways in the HPA axis under acute (ASDS) versus chronic social defeat stress (CSDS). Red bars denote pathways with significant up-regulation (*p* < 0.05). (**B**) Transcriptional profiles of ASDS-induced inflammatory pathways in the hypothalamus,
highlighting the activation of NF-κB, IL-17, ECM–receptor interaction, and TNF signaling pathways.
(**C**) Transcriptional profiles of CSDS-induced inflammatory pathways in the hypothalamus, highlighting
the activation of NF-κB, IL-17, ECM–receptor interaction, and TNF signaling pathways. (**D**) Comparative
analysis of neurotransmitter signaling pathways activity under ASDS versus CSDS across
hypothalamus, pituitary gland, and adrenal gland. Red bar indicates up-regulated pathways, while
blue bar indicates down-regulated pathways. (**E**) Gene expression profile of neurotransmitter-related
signaling pathways in the hypothalamus under ASDS. (**F**) Gene expression profile of neurotransmitterrelated
signaling pathways in the hypothalamus under CSDS. (**G**) A comparative analysis of hormone
signaling pathway activity under ASDS versus CSDS in the hypothalamus, pituitary gland, and
adrenal gland. (**H**) A gene expression profile of hormone-related signaling pathways in the pituitary
gland under ASDS. (**I**) A gene expression profile of hormone-related signaling pathways in
the pituitary gland under CSDS. (**J**) A depiction of ligand–receptor interactions in the HPA axis
under CSDS.

## Data Availability

Data are contained within the article or Appendix A. For other information, please contact the corresponding author.

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
