# Peer review of "Comparative Analysis of HPA-Axis Dysregulation and Dynamic Molecular Mechanisms in Acute Versus Chronic Social Defeat Stress"

_ijms, 2025, doi:10.3390/ijms26136063_

Round 1

Reviewer 1 Report

Comments and Suggestions for Authors
  1. The title and abstract should be revised. The term "stress" implies a very broad scope, which this study does not present as such. The authors should specify the scope to be "social defeat stress" or any relevant terms.
  2. There is a distinct difference between physiological (restraint, forced swimming, etc) and psychological (fear, social defeat, etc) stress. HOwever the introduction and are mainly focused on the stress in a broad aspect. The author should focus on social defeat stress.
  3.  There are too many abbreviations used in the manuscript, which serve NO PURPOSE apart from creating confusion among the readers. MDPI journals do not impose word limits. 
  4. The authors should be careful in labeling of treatment/experimental groups. In the figures, especially the bar graphs, the "Control" and "Sham" are in different groups. However, in the protocol section, line 446, as quoted "The control group C57BL/6 mice (Sham) underwent a sham surgery". This indicate that sham is the control group. It is very confusing and may create misinterpretation of the results. The reviewer is unable to be sure if the results presented is accurate. sometimies, the sham and control groups were used interchangeably in the article, which create doubts on the validity of the results. 
  5.  The average weight of the C57BL/6 mice is missing in th animals section.

Author Response

Comments 1: The title and abstract should be revised. The term "stress" implies a very broad scope, which this study does not present as such. The authors should specify the scope to be "social defeat stress" or any relevant terms.

Response 1: Thank you for the reviewer's valuable comments. Indeed, as pointed out, the term "stress" encompasses a broad range of conditions, whereas our study primarily focuses on a specific type of stress response-social defeat stress. Therefore, we have replaced the term "stress" with "social defeat stress" in both the title and main text to more accurately reflect the focus of our research and to help readers better recognize the unique perspective and contributions of this study (Page 1, Line 5-9).

Comments 2: There is a distinct difference between physiological (restraint, forced swimming, etc) and psychological (fear, social defeat, etc) stress. However, the introduction and are mainly focused on the stress in a broad aspect. The author should focus on social defeat stress.

Response 2: Thank you for the reviewer's valuable comments. We acknowledge that there is indeed a significant difference between physiological stress (such as restraint, forced swimming, etc.) and psychological stress (such as fear, social defeat, etc.). In accordance with your suggestion, we have reviewed the related literature on social defeat stress and conducted a detailed description in the introduction part (Page 1, Line 21-33, 50-73).

Comments 3: There are too many abbreviations used in the manuscript, which serve NO PURPOSE apart from creating confusion among the readers. MDPI journals do not impose word limits.

Response 3: Thank you for your valuable comment regarding the excessive use of abbreviations in the manuscript.

In response, we have carefully reviewed all abbreviations used in the text and made the following revisions: The abbreviation "ADX" has been expanded to "adrenalectomy" in all sections to ensure clarity throughout the revised manuscript. To avoid confusion for the readers, we have replaced the term 'CS' with 'CSDS' throughout the manuscript, and replaced the term 'AS' with 'ASDS' throughout the manuscript. We have retained only essential and widely accepted abbreviations (e.g., HPA axis, CSDS, GH, OFT, SIT), which are standard in the field and do not compromise readability. These changes aim to improve the overall clarity and accessibility of the manuscript for a broader readership (Page4, 6, Line 153-169,180-204).

Comments 4: The authors should be careful in labeling of treatment/experimental groups. In the figures, especially the bar graphs, the "Control" and "Sham" are in different groups. However, in the protocol section, line 446, as quoted "The control group C57BL/6 mice (Sham) underwent a sham surgery". This indicate that sham is the control group. It is very confusing and may create misinterpretation of the results. The reviewer is unable to be sure if the results presented is accurate. sometimies, the sham and control groups were used interchangeably in the article, which create doubts on the validity of the results.

Response 4: Thank you for your insightful comment regarding the labeling of experimental groups. We apologize for the ambiguity caused by the interchangeable use of "Control" and "Sham" in some parts of the manuscript. In our study, the Sham group (C57BL/6 mice) refers to the control group that underwent sham surgery---a procedure involving a small incision in the abdominal cavity followed by suturing without further intervention. This group was not exposed to any stress paradigm or treatment and served as the surgical control. To address this issue, we have carefully revised the text throughout the revised manuscript to ensure consistent terminology: In the revised version, we now uniformly refer to this group as the "Sham group", to clearly distinguish it from other experimental conditions. The sentence on line 446 has been rephrased as follows:"C57BL/6 mice were assigned to the sham group and underwent sham surgery, which consisted of making a small incision in the abdominal cavity followed by suturing without further intervention" (Page17, Line597-599).

Comments 5: The average weight of the C57BL/6 mice is missing in th animals section.

Response 5: Thanks. The average body weight (22-24 g) of the C57BL/6 mice has been included in the Materials and Methods part (Page15,Line 497).

Reviewer 2 Report

Comments and Suggestions for Authors

In this manuscript, Yang et al. report a comparative analysis of hpa axis dysregulation and dynamic molecular mechanisms in acute versus chronic stress. The Authors conclude that the study identifies potential therapeutic targets to mitigate chronic stress effects and offers a theoretical foundation for personalized interventions. The manuscript is interesting. However, some points should be addressed.

- The introduction is very short. In particular, the Authors must  provide more information about differences between acute and chronic stress.

- The major limitation of this study is the use of only male mice. The Authors must consider the stress-related disorders are more common in women rather than in men. There are updated CSDS models established also in female mice (PMID: 34587653). Moreover it is reported that the biological sex is a key factor leading to sex differences in the responses to acute and chronic stress (PMID: 37293561; PMID: 33867113). The Authors must at least discuss these crucial aspects.

- Line 78: The Authors wrote: “we developed experimental models for both stress conditions”. I guess the Authors used established models.

- Line 90: The Authors wrote: “In contrast, the chronic stress (CS-24 h) group exhibited persistent behavioral changes”. To write persistent behavioral changes, the Authors must show that these behavioral abnormalities must persist for weeks or months. Thus, the Authors must specify that these alterations are longer than those ones induced by acute stress.  

-Line 102: The Authors wrote: “Serum cortisol levels were measured post-adrenalectomy.” I guess the Authors wanted to write corticosterone considering this is a preclinical study involving rodents.

- The discussion needs to be improved. The Authors should better discuss the clinical relevance of the targets identified.

- The Authors must check the presence of typos throughout the manuscript.

- The Authors must check the presence of statements without references and insert the appropriate references.

Author Response

Comments 1: The introduction is very short. In particular, the Authors must  provide more information about differences between acute and chronic stress.

Response 1: Thank you for this valuable comment. In response, we have expanded the Introduction section by adding a detailed description of the physiological and molecular differences between acute and chronic stress in the revised manuscript (Page 2, Line 50-63). Furthermore, to avoid potential misinterpretation caused by the general term “chronic stress (CS)”, we have clarified in the text that all chronic stress conditions in our study specifically refer to chronic social defeat stress (CSDS). This change ensures greater precision and helps readers better understand the nature and scope of our experimental model. In addition, we have reviewed the related literature on social defeat stress and conducted a detailed description in the introduction part.

Comments 2: The major limitation of this study is the use of only male mice. The Authors must consider the stress-related disorders are more common in women rather than in men. There are updated CSDS models established also in female mice (PMID: 34587653). Moreover it is reported that the biological sex is a key factor leading to sex differences in the responses to acute and chronic stress (PMID: 37293561; PMID: 33867113). The Authors must at least discuss these crucial aspects. 

Response 2: We sincerely appreciate the reviewer’s insightful comment regarding the exclusive use of male mice in our study and the importance of considering sex differences in stress-related disorders, particularly given their higher prevalence in women. We fully agree that biological sex is a critical variable in the response to both acute and chronic stress.

There are several well-established animal models such as the chronic social defeat stress (CSDS) model, the chronic unpredictable mild stress (CUMS) model, chronic restraint stress model (CRS), and the learned helplessness model used to explore how stress affects physiology, immunity, and behavior in rodents. The CSDS model has a major advantage in that it closely mimics how social stress can induce anxiety- and depression-like behaviors. Therefore, we employed both acute and chronic social defeat stress paradigms to investigate the effects of social stress on the HPA axis and behavior.

As the reviewer pointed out, many effects of psychosocial stress are sex-dependent, and exploring the influence of stress on behavior and physiology across sexes is of great significance. However, CD1 aggressor mice typically display reduced aggression toward female intruders, resulting in lower model reliability when using female C57BL/6 mice. Therefore, the classic CSDS protocol has been predominantly established using male mice.   As noted by the reviewer, recent studies have developed modified protocols for female CSDS models. Furman et al. (Neurosci.; PMID: 345876531) introduced a novel chronic social stress paradigm in female mice by integrating multiple social stress components, including resident-intruder interactions, social crowding, and social instability. Additionally, several recent reports emphasize the role of sex differences under various stress paradigms (Neurobiol.Stress, PMID: 372935612; Biol.Psychiatry, PMID: 338671133). In particular, the social variable chronic stress (SVCS) model has been shown to be more suitable for detecting sex-specific responses to chronic stress, especially when comparing behavioral and neurobiological outcomes between males and females. Each animal model provides specific advantages for addressing different scientific questions.

Given that the central objective of our study was to compare the effects of acute versus chronic stress on physiological and behavioral outcomes, we based our acute stress protocol on two previously published studies (Nat.Protoc.,PMC32202784, subthreshold stress model; Cell, PMID: 370015005, initial stress stage). As a consistent subject choice, both models utilized male mice. To maintain model reproducibility and comparability, we selected the acute social defeat stress (ASDS) model and the classical chronic social defeat stress (CSDS) model for our study, both employing a male mouse social stress paradigm. In the future study, we plan to incorporate female chronic social stress models to better understand sex-dependent responses to stress and to further explore the neurobiological and behavioral mechanisms underlying acute versus chronic stress exposure. We have now included a detailed discussion of these crucial aspects in the Materials and Methods part of the revised manuscript (Page15, Line 510-531).

Comments 3: Line 78: The Authors wrote: “we developed experimental models for both stress conditions”. I guess the Authors used established models.

Response 3: We appreciate the reviewer's insightful comment. In our study, we employed both established and modified protocols to investigate different levels of social defeat stress.

For the chronic social defeat stress (CSDS) models, we strictly followed the experimental protocol established by Krishnan et al. (Cell, PMID: 179567386) and the well-characterized procedure described by Golden, Sam A et al. (Nat.Protoc., PMC32202784).

For the subthreshold stress model, we mainly based our protocol on the well-established method described by Golden, Sam A et al. (Nat.Protoc., PMC32202784), where C57BL/6J mice undergo three consecutive 5-minute defeat sessions in a single day with 15-minute rest intervals, followed by social interaction testing 24 hours later - a protocol that does not induce significant avoidance in wild-type mice. Additionally, we referenced the approach described by Li, Huan-Yu et al. (Cell, PMID: 370015005), which employs a single-stimulus paradigm similar to the initial stress stage of CSDS modeling.

In this study, to examine the behavioral and HPA axis transcriptional responses to milder acute social defeat stress and better mimic the stress responses experienced in human daily life, we made specific modifications to this established protocol: Male C57 mice experienced a single 5-minute social defeat session and then seperated a baffle with holes. Notably, our behavioral results at 24-hours post-defeat were consistent with those reported by Golden, Sam A et al. (Nat.Protoc., PMC32202784), validating the successful implementation of this modified model. These protocol adaptations allowed us to specifically investigate the acute phase responses to milder stress exposure while maintaining the core features of the established subthreshold stress paradigm. In the revised manuscript, we have included additional references related to model construction (Page16, Line 524-531).

Comments 4: Line 90: The Authors wrote: “In contrast, the chronic stress (CS-24 h) group exhibited persistent behavioral changes”. To write persistent behavioral changes, the Authors must show that these behavioral abnormalities must persist for weeks or months. Thus, the Authors must specify that these alterations are longer than those ones induced by acute stress. 

Response 4: We thank the reviewer for this insightful comment. In our original statement that “the chronic social defeat stress (CSDS-24 h) group exhibited persistent behavioral changes,” we intended to indicate that these changes were more enduring compared to those induced by acute stress. As the reviewer correctly points out, the term "persistent" should ideally refer to changes that last for weeks or even months. Supporting this interpretation, previous studies have demonstrated that mice subjected to chronic social defeat stress (CSDS) exhibit long-lasting social avoidance behavior that can persist for 39 days. Notably, stable social defeat behavior is typically established by day 11 in the CSDS model, and this phenotype can be maintained until at least day 39 (Response Table 1, Cell, PMID: 179567386; Cell, PMID: 370015005).
Response Table 1

Response Table 1. The construction of the CSDS model and the behavioral phenotypes on Day 11 and Day 39 (Cell, 2007).

In addition, in our study, we also observed that the social avoidance behavior induced by a 10-day chronic social defeat stress (CSDS) protocol persisted up to day 38 (Response Figure 1). In contrast, mice subjected to only one day of social defeat stress showed recovery to baseline levels within 24 hours (as shown in the main text, Response Figure 2). These findings further support the sustained and robust nature of the 10-day CSDS model in modeling chronic stress responses. We have included the Day 38 behavioral data from the CSDS model in the supplementary figure of the revised manuscript (Supplementary Figure1 A-B).
Response Figure 1

Response Figure 1.  The behavioral phenotypes of the CSDS model on Day 38 (Our Data)

Response Figure 2

Response Figure 2.  The behavioral phenotypes of the ASDS model within 24 hours (Our Data)

Comments 5: Line 102: The Authors wrote: “Serum cortisol levels were measured post-adrenalectomy.” I guess the Authors wanted to write corticosterone considering this is a preclinical study involving rodents.

Response 5: We sincerely thank the reviewer for pointing out this issue. Indeed, there was a mistake in our original manuscript. Cortisol is the primary glucocorticoid in humans and other primates, whereas in rodents, corticosterone is the major glucocorticoid and serves functions analogous to those of cortisol in humans. We have now corrected all instances of "cortisol" to "corticosterone" in the revised manuscript. (Page6, Line 178,183-187,195,208)

Comments 6: The discussion needs to be improved. The Authors should better discuss the clinical relevance of the targets identified.

Response 6: We appreciate the reviewer's suggestion and have strengthened the clinical discussion in the revised manuscript. Our findings demonstrate that chronic social defeat stress leads to sustained activation of the IL-17 pathway and upregulation of Lcn2, which are closely associated with clinical observations in patients with anxiety and depression. Notably, IL-17A has been proposed as a potential biomarker for antidepressant response, and anti-IL-17A therapies (e.g., Ixekizumab) have shown preliminary efficacy in reducing depressive symptoms in psoriasis patients. However, blockade of IL-17 signaling has also been linked to increased psychiatric risk, including suicidality, highlighting its complex role in mood regulation. Preclinical evidence further supports a pro-depressive role of IL-17A, consistent with our findings.

In addition, we observed chronic stress-induced imbalances between GABA and glutamate signaling, recapitulating key neurochemical alterations seen in MDD patients. These findings suggest that the molecular signatures identified in our study, particularly the IL-17–Lcn2 axis and GABA/glutamate dysregulation, may serve as promising diagnostic or therapeutic targets for stress-related disorders. (Page 14,15, Line 467,494)

Comments 7: The Authors must check the presence of typos throughout the manuscript.

Response 7: Thank you. We have thoroughly checked the manuscript for any typos and have made the necessary corrections. Additionally, we have carefully proofread the text to ensure clarity and accuracy throughout.

Comments 8: The Authors must check the presence of statements without references and insert the appropriate references.

Response 8: Thanks. We have added the missing reference in the Methods section of the revised manuscript. (Page 16, Line 528,546; Page 17, Line 571, 584,593)

Reference

  1. Furman, O., Tsoory, M. & Chen, A. Differential chronic social stress models in male and female mice. Eur. J. Neurosci. 55, 2777–2793 (2022).
  2. Torrisi, S. A. et al. Acute stress alters recognition memory and AMPA/NMDA receptor subunits in a sex-dependent manner. Neurobiol. Stress 25, 100545 (2023).
  3. Bittar, T. P. et al. Chronic Stress Induces Sex-Specific Functional and Morphological Alterations in Corticoaccumbal and Corticotegmental Pathways. Biol. Psychiatry 90, 194–205 (2021).
  4. Golden, S. A., Covington, H. E., Berton, O. & Russo, S. J. A standardized protocol for repeated social defeat stress in mice. Nat. Protoc. 6, 1183–1191 (2011).
  5. Li, H.-Y. et al. A thalamic-primary auditory cortex circuit mediates resilience to stress. Cell 186, 1352-1368.e18 (2023).
  6. Krishnan, V. et al. Molecular Adaptations Underlying Susceptibility and Resistance to Social Defeat in Brain Reward Regions. Cell 131, 391–404 (2007).

Round 2

Reviewer 1 Report

Comments and Suggestions for Authors

The authors have addressed my previous comments.

I have no more comments.

Reviewer 2 Report

Comments and Suggestions for Authors

The Authors have successfully addressed all the points I raised.